# The Influence of Ultrasonic Activation on Microstructure, Phase Transformation and Mechanical Properties of Porous Ni-Ti Shape Memory Alloys via Self-Propagating High-Temperature Synthesis

**DOI:** 10.3390/ma16186134

**Published:** 2023-09-09

**Authors:** Dovchinvanchig Maashaa, Enkhtsetseg Purevdagva, Vasili V. Rubanik, Vasili V. Rubanik

**Affiliations:** 1Department of Physics and Mathematics, School of Applied Sciences, Mongolian University of Life Sciences, Ulaanbaatar 17024, Mongolia; enkhtsetseg@muls.edu.mn; 2Institute of Technical Acoustics of National Academy of Science of Belarus, 210009 Vitebsk, Belarus; v.v.rubanik@tut.by (V.V.R.); jr@tut.by (V.V.R.J.)

**Keywords:** porous Ni-Ti SMA, microstructure, mechanical properties, phase transformation

## Abstract

Porous Ni-Ti shape memory alloys (SMAs) have been widely studied in biomedical and engineering applications. Porous Ni-Ti SMAs were obtained via self-propagating high-temperature synthesis (SHS), and their microstructure, phase transformation, and mechanical properties were investigated. This article presents the results of a study of changes in the microstructure, phase transformation, and mechanical properties of porous Ni-Ti SMAs when Ni and Ti metal powders were preliminarily subjected to ultrasonic activation at various periods. It was determined that the porosity of the obtained alloy samples was 62–68 vol%. The microstructure was composed of the main matrix Ni-Ti phase and the accompanying Ti and Ti_2_-Ni phases. The results show that the hardness 34.1–86.8 HB and elastic modulus 4.2–10.8 GPa increased with an increase in the ultrasonic activation time of the samples. The phase transformation temperature of the Ni-Ti shape memory alloy remained almost unchanged under the influence of ultrasonic treatment.

## 1. Introduction

Ni-Ti shape memory alloys (SMAs) are smart structural alloys with wide applications due to their excellent and unique mechanical properties, such as the shape memory effect (SME), superelasticity (SE), and corrosion resistance. Shape memory alloys are used in biomedical, robotics, automotive, and aerospace applications [1].

The recently obtained Ni-Ti SMA as a material with a porous structure, low density, and high porosity has attracted the interest of researchers due to its unusual mechanical properties, similar to some natural biomaterials [2].

Among the current hard-tissue implantable biomaterials, commercially pure titanium or titanium alloys are considered the best choice because of their good load-bearing capability, high specific strength, excellent corrosion resistance, and biocompatibility.

Porous Ni-Ti SMA has been applied successfully in medical fields, such as biomedical implants (e.g., orthopedics, traumatology, vertebrarium, maxillofacial surgery, and so on), and it has also been used as a medical instrument for cryogenic applications. The application field of porous Ni-Ti SMA is expanding now.

Porous Ni-Ti SMAs, due to their porous structure similar to the structure of human bones, have the properties of supporting the metabolism of living organisms, cell growth, and improving the ligamentous apparatus of the joints, and are used as rigid implants for teeth, bones, muscle tissue, artificial organs, patching holes, and stenting heart [3,4].

Porous Ni-Ti SMAs have been prepared using the diffusion sintering method [5], powder metallurgy [6], microwave sintering [7], and self-propagating high-temperature synthesis (SHS). Obtaining Ni-Ti alloys via the self-propagating high-temperature synthesis (SHS) method is an efficient and important method that saves energy and time compared to conventional techniques [8]. Factors affecting the properties of the alloys, such as the density, heating temperature, and heating rate, have been identified by many researchers.

Ni-Ti SMAs obtained via self-propagating high-temperature synthesis contain many secondary phases. Although porous Ni-Ti SMAs contain secondary phases of Ni-Ti_2_ and Ni_3_-Ti, which impart brittleness to the material and increase the risk of corrosion, these alloys have a very low corrosion rate compared to other biomaterials [9].

The most appropriate solution to this problem is heat-treatment research to change the material strength and phase temperature. Researchers have developed a heat-treatment method to obtain porous pure Ni-Ti SMAs [9].

Research has been carried out to change the microstructure and improve the mechanical properties of Ti-Al and Al-Mg_3_ alloys [10,11] and the shape memory effect via temperature variation or ultrasonic vibrations in the ultrasonic treatment of Ni-Ti SMAs [12]. Previous studies have shown that the thermal conductivity of Ni-Ti powder can be changed via ultrasonic vibrations [13]. Ultrasonic activation resulted in the formation of a more isotropic porous structure, which was formed during SHS, due to a change in the mode of the propagation of the combustion wave from pulsed to stationary. Ultrasonic vibration treatment reduces the size and volume fraction of precipitates and makes the chemical composition of the Ni-Ti phase more uniform, which affects the phase transformation and alloy characteristics [14].

The purpose of this work is to study the influence of the preliminary activation of metal powders using ultrasound at different periods depending on the activation time on the microstructure, porosity, phase transformation, and mechanical properties of Ni-Ti SMAs by SHS.

## 2. Materials and Methods

The powders of high-purity Ti (catalog type PTOM-2) and Ni (catalog type PNK-1L5) with an average particle size of 40 μm were mixed in the composition of Ti 50% and Ni 50%. The powders were preliminarily kept in an oven at a temperature of 100 °C for 4 h. The mixture was prepared in the ratio Ti–50.0 at.% Ni with a total weight of 200 g. Titanium and nickel powders combined in this proportion were mixed for 8 h in a V-shaped mixer with a rotation frequency of 0.5 Hz.

The samples were activated using the ultrasonic disperser UZDN-1M with a frequency of 22 kHz and an amplitude of 25 μm for 0, 15, 30, 60, and 120 min. To stabilize the treatment temperature, the working volume was thermostatically controlled by cooling it with running water.

The mixture was placed into a quartz tube with a diameter of 30 mm (mixture green density was 2.4 g/cm^3^) inside the thermal chamber, heated to 350 °C in an argon atmosphere (p = 1 atm), and held for 1 h. Then, the SHS reaction was initiated by heating the tungsten wire using an electrical current. The SHS reaction took place for 5 s, and then the samples were cooled inside the thermal chamber. SEM observations were conducted using a Jeol-6000Plus instrument (Jeol, Akishima, Japan) via the SEM method equipped with energy-dispersive spectroscopy (EDS) analysis systems made by Oxford. The hardness was measured with an HBE-3000A Electronic Brinell Hardness Tester by pressing a ball 10 mm in diameter with a force of 3000 N, and the average value of 10 measurements was obtained. Elasticity was measured on a microcomputer-controlled tensile testing machine CMT2503 on specimens 6 mm in diameter and 24 mm in length at a speed of 0.3 mm/min. Heat flux measurements were carried out during cooling and heating using a DSC 214 Polyma differential scanning calorimeter (NETZSCH-Gerätebau GmbH). The rate of cooling and heating was 10 °C/min, and the temperature range of the studies was from −60 °C to 140 °C. Proteus software was used to process the results.

## 3. Results and Discussion

### 3.1. The Porosity of the Ni-Ti SMA

The porosity of the obtained Ni-Ti SMAs was calculated using the following Formula (1) [14,15]:(1)$$P=(1-\frac{\rho }{\rho_{0}})\times 100\%$$
where

*ρ* is the density, calculated based on the mass and volume values of the Ni-Ti porous alloy;

*ρ*_0_ is the theoretical density of the material (6.21 g⁄cm^3^).

The porosity of the obtained Ni-Ti SMAs was 62.25–68.27 vol%, as shown in Table 1.

If the amount of porosity (50–70%) is high, the open-porosity Ni-Ti SMAs exhibit shape memory behavior as observed in porous Ni-Ti shape memory alloys [13,15].

### 3.2. Microstructure

The results of the surface morphology measurements of the obtained samples are shown in Figure 1. The measurements were carried out in the image plane at an accelerating voltage of 15 kV, the distance from the electron generator to the specimen was 5 mm, and the measurement scale was from 10 µm to 1 mm. As can be seen from the microstructure image of the lateral surface of the sample (Figure 1a), the shape of the pores of the untreated sample was a uniform channel with an average width of about 250 µm and a length of about 1000 µm. The pore size of the samples treated with ultrasound was larger, 500–1000 μm, and the shape became mostly round. According to the results of measurements of the surface morphology image on a scale of 10 μm, in general, a gray phase main matrix and black phases formed on the surface of the alloy.

As can be seen from the results of the porous Ni-Ti SMA ultrasonic activation for 120 min, the hollow part of the alloy is reduced, which is shown in Figure 1e. Ultrasonic action before the SHS process may produce a uniform structure without cavities in the sample. The number of precipitates on the surface decreased, and the Ni-Ti fraction increased as the duration of ultrasound activation increased Figure 1.

The composition of the phases was measured by EDS. The elemental composition of the porous Ni-Ti SMAs is shown in Table 2. In all porous Ni-Ti SMAs in the gray matrix phase, the volume ratio of Ti-Ni was approximately 1:1, so the matrix phase was the Ni-Ti phase. In the porous Ni-Ti SMAs obtained using the method of self-propagating high-temperature synthesis, various additional phases were formed, such as Ti2Ni, Ni3Ti, and Ni4Ti3. [14]. On the surface of the non-ultrasound-treated porous Ni-Ti SMAs, the black phase was formed only by the Ti phase (Figure 2a). The black phase formed on the surface of the sample subjected to ultrasonic treatment for 15 min also represented the Ti phase (Figure 2b). The black phase formed on the surface of the samples treated with ultrasound for 30 min, 60 min, and 120 min was the Ti_2_-Ni phase, and the Ti: Ni ratio was approximately 2:1 (Figure 2c–e).

### 3.3. Mechanical Properties

Then, the diameter d of the spherical depression created by the hardness tester ball was determined, and the hardness of the alloy was calculated according to the following Formula (2):(2)$$HB=\frac{2P}{\pi D(D-\sqrt{D^{2}-d^{2}})}$$

The cylindrical samples have a diameter of 1.4 cm and a height of 2.4 cm. The diameter of the pressing ball was *D* = 10 mm and the pressing force was *P* = 3000 N. The hardness measurement results are shown in Table 3. According to the test results, the hardness of the alloy increases with an increase in the ultrasonic treatment time of the sample mixture. The hardness of the sample without ultrasonic treatment is 34.1 HB, while that of the sample treated for 120 min is 86.8 HB (Figure 3). The increase in alloy hardness due to ultrasonic treatment corresponds to the results of research on Ti-Al and Al-Mg alloys [10,11].

The elastic modulus measurement results show that the elastic modulus of the porous Ni-Ti SMAs is 4.2–10.8 GPa. The elastic modulus of the sample not treated with ultrasound is 4.2 GPa, whereas the sample treated with ultrasound for 120 min is 10.8 GPa. When the Ni/Ti mixture is exposed to ultrasound, the elastic modulus of the Ni-Ti alloys increases (Figure 3). The mechanical properties of such porous materials depend greatly on the pore size, edges, shape, porosity, microstructure, and phase composition. In terms of the microstructure, it can be seen from Figure 1 that the pore shape of the non-ultrasonic sample is a uniform channel, whereas the pore shape of the ultrasonically treated sample is generally circular. It can be seen from Figure 2 that the amount of Ni-Ti phase increases with an increase in the ultrasound activation time. In addition, the mechanical properties are improved because the number of voids in the porous Ni-Ti SMAs is reduced. This result reaches the elastic modulus of the human bone (3–20 GPa) for the cortical layer of the bone [16]. High open porosity allows one to decrease the elastic modulus from 40–70 GPa, which is typical for bulk Ni-Ti alloys, to 3–20 GPa, which is typical for bone tissue. The high open-porosity Ni-Ti alloy makes it possible to reduce the elastic modulus of the bulk Ni-Ti alloy from 40–70 to 3–20 for use in bone tissue.

### 3.4. Phase Transformation

Typical calorimetric curves for a sample obtained using the SHS method are shown in Figure 4. The samples were activated using the parameters of ultrasonic (Table 4).

Figure 4 shows two calorimetric peaks for all porous samples obtained via SHS upon cooling and two peaks upon heating. The high-temperature peaks are due to the B2→B19’ transformation in Ti-rich Ni-Ti and the low-temperature peaks are due to the transformation in Ni-rich Ni-Ti. The low intensity of the high-temperature peaks of sample M1 indicates that the volume fraction of the Ti-rich Ni-Ti regions is smaller than the fraction of the Ni-rich Ni-Ti regions [14,17]. According to the DCS results, a minor Ni-rich Ni-Ti phase may have formed on the microstructure.

The preliminary ultrasonic treatment increases the intensity of the high-temperature peaks. The longer the duration and amplitude of the ultrasonic testing, the greater the intensity of the peaks observed. This can be explained by an increase in the volume fraction of the Ti-rich Ni-Ti phase. This indicates that the volume fraction of the Ti-rich Ni-Ti phase increases when the contact of the Ti and Ni powders with each other is activated due to ultrasound.

The characteristic temperatures of the low-temperature and high-temperature peaks practically do not change (Table 5), which indicates a constant Ti concentration in the Ni-Ti regions subjected to such transformations. This may be due to the uniform distribution of the Ti_2_Ni precipitates.

## 4. Conclusions

In this study, a Ni-Ti shape memory alloy with a porous structure was obtained using the method of self-propagating high-temperature synthesis (SHS). The samples were prepared for synthesis by preliminarily subjecting them to ultrasonic treatment by changing the activation time. The influence of the time of exposure to ultrasound preactivation on the microstructure, phase transition, and mechanical properties was studied. The porosity of the obtained Ni-Ti SMAs was 62.25–68.27 vol%. Based on the results of this study, the following conclusions can be drawn:From the point of view of the microstructure, the shape of the pores of the sample not activated via ultrasound sample was uniformly channel-shaped with an average width of about 250 µm and a length of about 1000 µm. In the samples treated with ultrasound, structural changes occurred, the pore size became larger (500–1000 µm), and the shape became mostly round.The main matrix Ni-Ti phase and precipitates Ti and Ni-Ti_2_ are formed.The phase transformation temperature of the Ni-Ti shape memory alloy is almost unchanged under the influence of ultrasonic treatment.The results of the analysis of the mechanical properties of the alloy show that with an increase in the time of ultrasonic treatment, the hardness and elasticity increase. The hardness of the sample without ultrasonic treatment is 34.1 HB, whereas the sample treated for 120 min is 86.8 HB, and the elastic range is 4.2–10.8 GPa.

## Figures and Tables

**Figure 1 materials-16-06134-f001:**
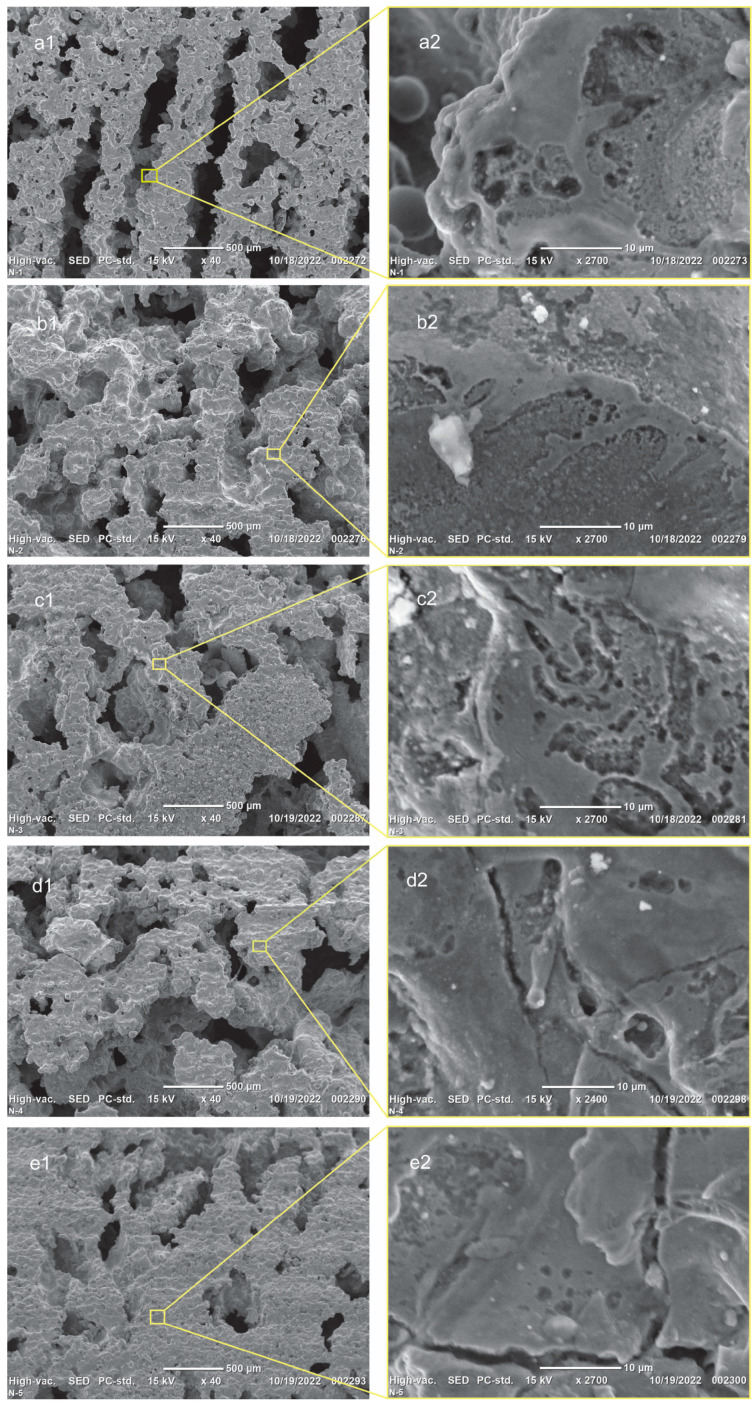
Micrographs of the lateral surface of porous Ni-Ti SMAs: (**a1**,**a2**) not treated with ultrasound; (**b1**,**b2**) ultrasound treated for 15 min; (**c1**,**c2**) ultrasound treated for 30 min; (**d1**,**d2**) ultrasound treated for 60 min; (**e1**,**e2**) ultrasound treated for 120 min.

**Figure 2 materials-16-06134-f002:**
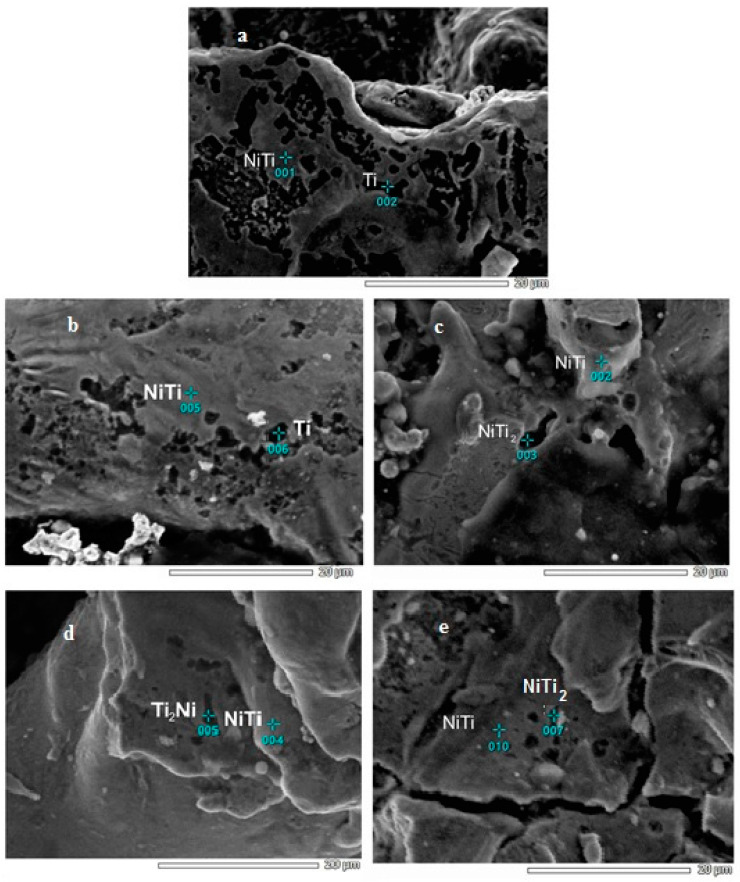
Microstructure and EDS results of porous Ni-Ti SMAs: (**a**) not treated with ultrasound; (**b**) ultrasound treated for 15 min; (**c**) ultrasound treated for 30 min; (**d**) ultrasound treated for 60 min; (**e**) ultrasound treated for 120 min.

**Figure 3 materials-16-06134-f003:**
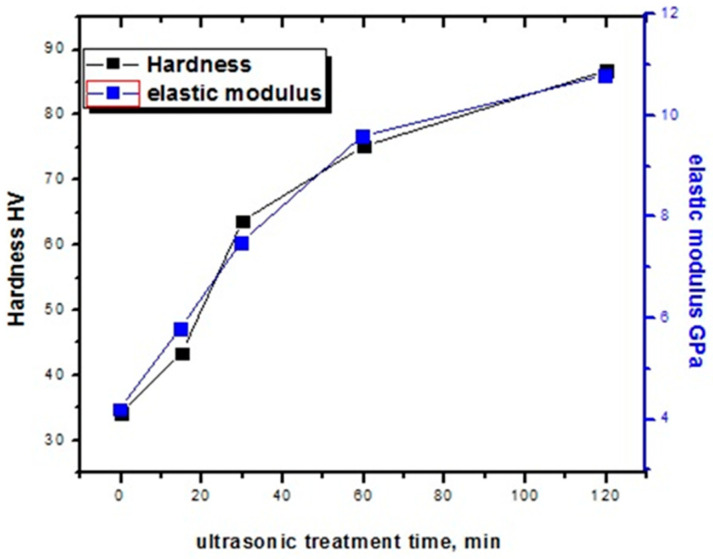
Hardness and elasticity modulus of the porous SMAs, depending on the time of ultrasonic treatment.

**Figure 4 materials-16-06134-f004:**
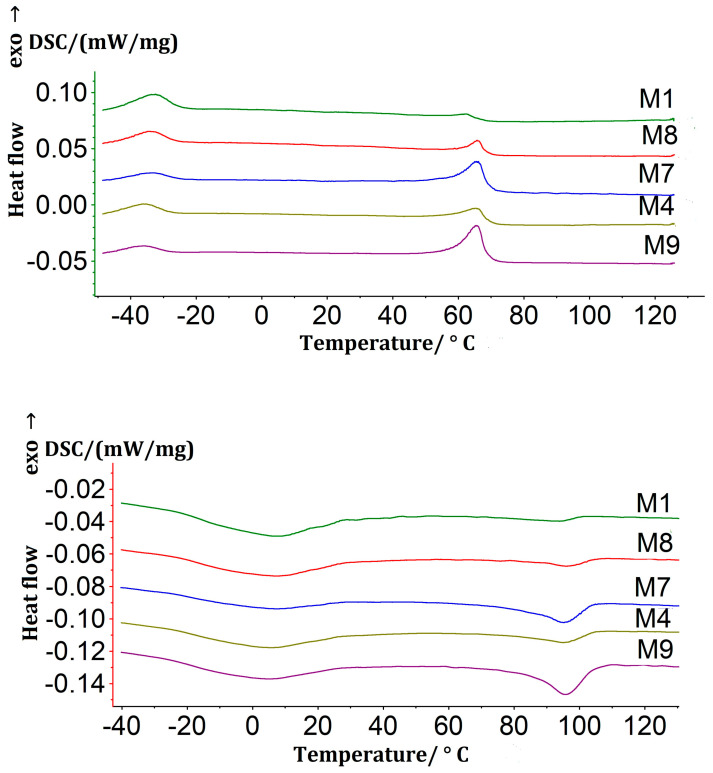
Calorimetric curves obtained during cooling and heating of porous Ni-Ti samples obtained by SHS without ultrasonic activation (M1) and after ultrasonic activation with an oscillation amplitude of 25 μm for 15 (M8), 30 (M7), 60 (M4), and 120 (M9) min.

**Table 1 materials-16-06134-t001:** The porosity of the Ni-Ti SMA samples.

№	Sample	Duration of Ultrasound Treatment, t (min)	Theoretical Den sity, *ρ*_t_ (g/cm^3^)	Sample Mass, (g)	Sample Volume, (cm)^3^	Sample Density, *ρ* (g/cm^3^)	Porosity Volume Ratio (%)
1	A	0	6.21	6.68	3.38	1,97	68.27
2	B	15	6.21	7.41	3.69	2	67.79
3	C	30	6.21	7.20	3.38	2.13	65.69
4	D	60	6.21	4.96	2.30	2.15	65.27
5	E	120	6.21	9.39	4	2.34	62.32

**Table 2 materials-16-06134-t002:** The composition of the Ni-Ti alloys.

№	Sample	Duration of Ultrasound Treatment, t (min)	Phase	Ti (at. %)	Ni (at. %)	Formed Phase
1	A	0	Matrix	57.13	42.87	Ni-Ti
			Black Phase	100	-	Ti
2	B	15	Matrix	54.02	45.98	Ni-Ti
			Black Phase	100	-	Ti
3	C	30	Matrix	54.82	45.18	Ni-Ti
			Black Phase	63.70	36.30	Ni-Ti2
4	D	60	Matrix	56.22	43.78	Ni-Ti
			Black Phase	71.92	28.08	Ni-Ti2
5	E	120	Matrix	53.03	46.97	Ni-Ti
			Black Phase	63.71	36.29	Ni-Ti2

**Table 3 materials-16-06134-t003:** The hardness of porous Ni-Ti SMA samples.

№	Sample	Duration of Ultrasound Treatment, t (min)	Diameter of the Depression, Aver d (mm)	Hardness
1	A	0	6.90	34.1 ± 1.2 HB
2	B	15	6.12	43.4 ± 0.8 HB
3	C	30	5.23	63.7 ± 1.5 HB
4	D	60	4.80	75.2 ± 1.2 HB
5	E	120	4.50	86.8 ± 1.8 HB

**Table 4 materials-16-06134-t004:** Parameters of ultrasonic treatment of the Ni/Ti mixture.

№	Sample Number	Amplitude, (µm)	Time, (min)	Frequency, (kHz)
1	M1	0	0	0
2	M4	25	60	22
3	M7	25	30	22
4	M8	25	15	22
5	M9	25	120	22

**Table 5 materials-16-06134-t005:** The phase transformation of porous Ni-Ti SMA samples.

Sample Number	M_s1_, °C	M_f1_, °C	A_s1_, °C	A_f1_, °C	M_s2_, °C	M_f2_, °C	A_s2_, °C	A_f2_, °C
M1	0	0	87	105	−25	−45	−22	27
M4	25	60	86	105	−27	−54	−24	26
M7	25	30	85	104	−25	−46	−15	25
M8	25	15	88	107	−27	−45	−22	27
M9	25	120	86	105	−28	−54	−24	27

## Data Availability

All the data are available in the main text.

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
