# Peer review of "The Influence of Ultrasonic Activation on Microstructure, Phase Transformation and Mechanical Properties of Porous Ni-Ti Shape Memory Alloys via Self-Propagating High-Temperature Synthesis"

_materials, 2023, doi:10.3390/ma16186134_

Round 1

Reviewer 1 Report

Dear authors,

The topic you are going to present is interesting and worth to be shared with the scientific community. However, before publication, your work needs extensive revisions.

1. Abstract and introduction are not very clear, part due to the English grammar, part to the clarity of the message you'd like to bring to the audience.

2. The experimental session can be improved by introducing more experimental details.

3. Results must be described in more detail, in particular the SEM pictures and EDS analysis. I was surprised you did not check the chemistry of SHS samples. Bulk elements and gas impurities are of key importance for NiTi. Moreover, you described the method to measure the elastic modulus, but you reported only the value in the graph. It should be useful (in particular for NiTi alloy) to the reader looking at the stress-strain curves to see the superelastic behavior of samples.

4. DSC curves demonstrate a general inhomogenity of your samples. It looks like the material has a two phase transformation because you have different chemistries after SHS process. This is not good for a potential orthopedic application. Please, clarify these aspects.

Thank you.

The English language is quite poor. It must be improved extensively within the overall work.

Author Response

Dear editor and reviewer:

Thank you very much for your reviewing. We revised our manuscript according to your comments. The comments and our responses are as follows:

The topic you are going to present is interesting and worth to be shared with the scientific community. However, before publication, your work needs extensive revisions.

  1. Abstract and introduction are not very clear, part due to the English grammar, part to the clarity of the message you'd like to bring to the audience.

Response: We have been improvements to the abstract and introduction.

We rechecked the text several times and tried our best to improve the English. If the English still cannot meet the criteria for publication, we will further ask the language editor for help.

  1. The experimental session can be improved by introducing more experimental details.

Response: We believe that the experimental details have been included.

  1. Results must be described in more detail, in particular the SEM pictures and EDS analysis. I was surprised you did not check the chemistry of SHS samples. Bulk elements and gas impurities are of key importance for NiTi. Moreover, you described the method to measure the elastic modulus, but you reported only the value in the graph. It should be useful (in particular for NiTi alloy) to the reader looking at the stress-strain curves to see the superelastic behavior of samples.

Response: All samples have been observed Jeol-6000Plus scanning electron microscope and EDS. Please see Fig 1 and table 2. A graph was drawn comparing the amount of elastic modulus with the amount of hardness.

  1. DSC curves demonstrate a general inhomogenity of your samples. It looks like the material has a two phase transformation because you have different chemistries after SHS process. This is not good for a potential orthopedic application. Please, clarify these aspects.

Response: Two calorimetric peaks of heat release were found on cooling and two peaks of heat absorption were observed on heating of the samples manufactured at a pre-heating temperature of 350°C. high-temperature two peaks were caused by the B2 ←→ B19’ transformation in the Ni-Ti areas with a Ti concentration of 50.0 at.% or more. Low-temperature two peaks were due to the same transformation occurred in the Ni-Ti phase with a Ni concentration larger than 50.5 at.%. An increase in the pre-heating temperature decreased the intensity of high-temperature peaks both on cooling and heating. The square under the calorimetric peak was proportional to the volume fraction of the alloy undergoing the martensitic transformation.

Reviewer 2 Report

see attached.

see attached.

Author Response

Dear editor and reviewer:

Thank you very much for your reviewing. We revised our manuscript according to your comments. The comments and our responses are as follows:

Changes in the microstructure, phase transformation of porous NiTi SMAs where Ni and Ti metal powders were preliminarily subjected to ultrasonic activation at various periods were investigated in this work. NiTi shape memory alloy with a porous structure by self-propagating high-temperature synthesis. Characterization of samples demonstrated the influence of ultrasonic treatment on their hardness and elasticity. In contrast, ultrasonic treatment has little influence in the phase transformation temperature. However, the following comments need to be considered carefully prior to consideration of publication.

  1. The title format of Section 3.1 is inconsistent with the rest.

Response: The title format of Section 3.1 has been edited.

  1. The sample density of Sample A in Table 1 seems incorrect. Please double check it.

Response: The sample density of Sample A in Table 1 was checked.

  1. It is mentioned in Section 3.2 that “The composition of the phases was measured by EDS. The elemental composition of porous Ni-Ti SMAs is shown in Table 1”. Double check Table 1. Is it should be replaced by Table 2?

Response: the number of table 2 has been changed to table 1.

  1. In Section 3.3 and Section 3.4, many parts of the text have no space in the first line.

Response: first line of text is spaced.

  1. The EDS result shown in Figs. 3~7 is very vague. Please double check.

Response: The EDS result is checked.

  1. 11 and 12 are of low quality. Please redraw them.

Response: Figs. 11 and 12  have been redrawn.

  1. There is no deep discussion on the results of characterization. How convincing are the results? What is the most remarkable point and creation in this research? Are there any disadvantages that can be taken care of in future research?

Response: The results of the analysis of the mechanical properties of the alloy show that with an increase in the time of ultrasonic treatment, the hardness and elasticity increase. The hardness of the sample without ultrasonic treatment is 34.1HB, while the sample treated for 120 min is 86.8HB, and the elastic range is 4.2-10.8 GPa. We have succeeded in improving the mechanical properties of the material by ultrasonic activation.

Reviewer 3 Report

The authors have studied the effects of ultrasonic activation on the microstructure, phase transformation, and mechanical properties of Nitinol. They have succeeded in improving the mechanical properties of the material by ultrasonic activation. The novelty of the work, the sufficiency of the characterizations, and the accuracy of the discussion meet the standards of the Journal. I support publishing the manuscript in MDPI Materials. I only have some comments to improve the manuscript:

1.       Title: missing comma “Microstructure, Phase Transformation”, what is “SHS” stand for?

2.       Abstract: “Porous Ni-Ti shape memory alloys (SMAs) have been widely studied in biomedical applications because they have a porous structure similar to human bone.” Not only! They have been widely studied in bio due to several things including this resemblance.  

3.       Abstract: “The results of show that” unclear

4.       Line 36: “Porous NiTi SMAs have been prepared methods diffusion sintering method” unclear sentence

5.       Lines 60-63: Too long sentence!

6.       What is the significance of reporting Figure 2? Can’t the images be added to Fig. 1 with the labels of the composition of different regions (As shown in Fig.3 to Fig. 7), all in one figure?

7.       Line 128: Fig. 2 or Fig. 3?

8.       No need for Fig.3 to Fig.7. They are more suitable for the supplementary file. Table 2 clearly reports all the essential numbers. Please also provide the amount of uncertainty in the EDS measurements.

9.       No need for Fig.10 to Fig.12. They are more suitable for the supplementary file. Table 5 clearly reports all the essential numbers.   

General comment: Up to section “3.4. Phase Transformation” everything is based on the time of sonication (Samples A-E) which is crystal clear. In section 3.4., the amplitude of the sonication is also introduced as a variable, and only the transformations are reported. How about the morphology and composition characterizations? In the abstract and conclusion also, there is no word on the effects of the amplitude. If the effects of the amplitude of the sonication were also studied, all the characterizations should be reported and the results should be highlighted in the abstract and conclusion. If not, better to exclude them because reporting those just in section 3.4. only disturbs the clarity and logic of the work.

Question: Did the authors also consider the structural characterization of the samples by EBSD or XRD?

Minor editing of the English language is required

Author Response

Dear editor and reviewer:

Thank you very much for your reviewing. We revised our manuscript according to your comments. The comments and our responses are as follows:

The authors have studied the effects of ultrasonic activation on the microstructure, phase transformation, and mechanical properties of Nitinol. They have succeeded in improving the mechanical properties of the material by ultrasonic activation. The novelty of the work, the sufficiency of the characterizations, and the accuracy of the discussion meet the standards of the Journal. I support publishing the manuscript in MDPI Materials. I only have some comments to improve the manuscript:

  1. Title: missing comma “Microstructure, Phase Transformation”, what is “SHS” stand for?

Response: The title has been edited.

  1. Abstract: “Porous Ni-Ti shape memory alloys (SMAs) have been widely studied in biomedical applications because they have a porous structure similar to human bone.” Not only! They have been widely studied in bio due to several things including this resemblance.  

Response: Detailed in the introduction section.

  1. Abstract: “The results of show that” unclear.

Response: Results are quantified in the abstract.

  1. Line 36: “Porous NiTi SMAs have been prepared methods diffusion sintering method” unclear sentence.

Response: General methods of sample preparation are included.

  1. Lines 60-63: Too long sentence.

Response: Corrected sentence.

  1. What is the significance of reporting Figure 2? Can’t the images be added to Fig. 1 with the labels of the composition of different regions (As shown in Fig.3 to Fig. 7), all in one figure?

Response: Figure 1 shows the general surface porosity.

Figure 2 shows the surface phase difference. All the pictures are too small to fit together, so I have included them separately.

  1. Line 128: Fig. 2 or Fig. 3?

Response: Fig.3 edited.

  1. No need for Fig.3 to Fig.7. They are more suitable for the supplementary file. Table 2 clearly reports all the essential numbers. Please also provide the amount of uncertainty in the EDS measurements.

Response: Figures 3-7 show surface phase differences and EDS results, as well as phase sizes.

  1. No need for Fig.10 to Fig.12. They are more suitable for the supplementary file. Table 5 clearly reports all the essential numbers.

Response: Fig 10 and Fig 12 has been removed.

Round 2

Reviewer 1 Report

Thanks for the extensive revision work.

Author Response

Dear editor and reviewer:

Thank you very much for your reviewing. We revised our manuscript according to your comments. 

Reviewer 3 Report

The authors have considered most of the comments and suggestions but still, there are a few comments that have not been sufficiently addressed. I suggest addressing the following comments and then the manuscript would be ready to be published:

2. Abstract: “Porous Ni-Ti shape memory alloys (SMAs) have been widely studied in biomedical applications because they have a porous structure similar to human bone.” Not only! They have been widely studied in bio due to several things including this resemblance.  

Response: Detailed in the introduction section.

In addition to the detailed explanations in the Introduction, it would be better if the authors also modify the above sentence in the Abstract so that it won't be misleading.

6. What is the significance of reporting Figure 2? Can’t the images be added to Fig. 1 with the labels of the composition of different regions (As shown in Fig.3 to Fig. 7), all in one figure?

Response: Figure 1 shows the general surface porosity. Figure 2 shows the surface phase difference. All the pictures are too small to fit together, so I have included them separately.

I suggest merging Figures 1 and 2. Each of the images in Figure 2 can be added to Figure 1 as an inset highlining the locations where the images were magnified.

8. No need for Fig.3 to Fig.7. They are more suitable for the supplementary file. Table 2 clearly reports all the essential numbers. Please also provide the amount of uncertainty in the EDS measurements.

Response: Figures 3-7 show surface phase differences and EDS results, as well as phase sizes.

I suggest merging Figures 3-7 into one Figure. To do that, some unnecessary graphs and images can be moved into the supplementary file. For instance, the graphs showing peaks of elements and numbers can be removed. To show the composition of each of the samples, one SEM image per sample is enough. The author may mark different regions of the image and write the measured composition on the same image. So it would become a Figure with 5 images (one/sample) with measured quantitative compositions (the composition of the dark region and the composition of the bright region).

General comment: Up to section “3.4. Phase Transformation” everything is based on the time of sonication (Samples A-E) which is crystal clear. In section 3.4., the amplitude of the sonication is also introduced as a variable, and only the transformations are reported. How about the morphology and composition characterizations? In the abstract and conclusion also, there is no word on the effects of the amplitude. If the effects of the amplitude of the sonication were also studied, all the characterizations should be reported and the results should be highlighted in the abstract and conclusion. If not, better to exclude them because reporting those just in section 3.4. only disturbs the clarity and logic of the work.

Question: Did the authors also consider the structural characterization of the samples by EBSD or XRD?  

Author Response

Dear editor and reviewer:

Thank you very much for your reviewing. We revised our manuscript according to your comments. The comments and our responses are as follows:

The authors have considered most of the comments and suggestions but still, there are a few comments that have not been sufficiently addressed. I suggest addressing the following comments and then the manuscript would be ready to be published:

  1. Abstract: “Porous Ni-Ti shape memory alloys (SMAs) have been widely studied in biomedical applications because they have a porous structure similar to human bone.” Not only! They have been widely studied in bio due to several things including this resemblance.  

Response: Detailed in the introduction section.

In addition to the detailed explanations in the Introduction, it would be better if the authors also modify the above sentence in the Abstract so that it won't be misleading.

Response: This sentence has been changed in general.

  1. What is the significance of reporting Figure 2? Can’t the images be added to Fig. 1 with the labels of the composition of different regions (As shown in Fig.3 to Fig. 7), all in one figure?

Response: Figure 1 shows the general surface porosity. Figure 2 shows the surface phase difference. All the pictures are too small to fit together, so I have included them separately.

I suggest merging Figures 1 and 2. Each of the images in Figure 2 can be added to Figure 1 as an inset highlining the locations where the images were magnified.

Response: We combined Figure 1 and Figure 2.

  1. No need for Fig.3 to Fig.7. They are more suitable for the supplementary file. Table 2 clearly reports all the essential numbers. Please also provide the amount of uncertainty in the EDS measurements.

Response: Figures 3-7 show surface phase differences and EDS results, as well as phase sizes.

I suggest merging Figures 3-7 into one Figure. To do that, some unnecessary graphs and images can be moved into the supplementary file. For instance, the graphs showing peaks of elements and numbers can be removed. To show the composition of each of the samples, one SEM image per sample is enough. The author may mark different regions of the image and write the measured composition on the same image. So it would become a Figure with 5 images (one/sample) with measured quantitative compositions (the composition of the dark region and the composition of the bright region).

Response: We combined pictures 3-7 into one picture. The EDS results graph has been removed. in our opinion, an EDS graphs would have been a more reliable result.

General comment: Up to section “3.4. Phase Transformation” everything is based on the time of sonication (Samples A-E) which is crystal clear. In section 3.4., the amplitude of the sonication is also introduced as a variable, and only the transformations are reported. How about the morphology and composition characterizations? In the abstract and conclusion also, there is no word on the effects of the amplitude. If the effects of the amplitude of the sonication were also studied, all the characterizations should be reported and the results should be highlighted in the abstract and conclusion. If not, better to exclude them because reporting those just in section 3.4. only disturbs the clarity and logic of the work.

Response: We removed the measurement result by changing the amplitude value. The amplitude value is chosen to be only 25 μm.

Question: Did the authors also consider the structural characterization of the samples by EBSD or XRD?  

Response: XRD measurements of some powder materials were performed. The results are not included in this article. It is consistent with the results of SEM.

Figure 1 – Initial powders of titanium (a) and nickel (b), their diffraction patterns (c)
